# Detection of Novel Biomarkers in Pediatric Autoimmune Hepatitis by Proteomic Profiling

**DOI:** 10.3390/ijms24087479

**Published:** 2023-04-19

**Authors:** Claudia Sîrbe, Medeea Badii, Tania O. Crişan, Gabriel Bența, Alina Grama, Leo A. B. Joosten, Simona Rednic, Tudor Lucian Pop

**Affiliations:** 12nd Pediatric Discipline, Department of Mother and Child, “Iuliu Hațieganu” University of Medicine and Pharmacy, 400012 Cluj-Napoca, Romania; claudia.sirbe@yahoo.com (C.S.);; 22nd Pediatric Clinic, Center of Expertise in Pediatric Liver Rare Disorders, Emergency Clinical Hospital for Children, 400177 Cluj-Napoca, Romania; 3Department of Medical Genetics, “Iuliu Hațieganu” University of Medicine and Pharmacy, 400012 Cluj-Napoca, Romania; 4Department of Internal Medicine, Radboud University Medical Centre, 6525 XZ Nijmegen, The Netherlands; 5Rheumatology Department, Emergency County Hospital Cluj, 400347 Cluj-Napoca, Romania; 6Rheumatology Discipline, “Iuliu Hațieganu” University of Medicine and Pharmacy, 400012 Cluj-Napoca, Romania

**Keywords:** autoimmune hepatitis, autoimmune sclerosing cholangitis, novel biomarkers, proteomic profiling, immune tolerance, hepatic inflammation, liver fibrosis

## Abstract

Autoimmune hepatitis (AIH) is characterized by immune-mediated hepatocyte injury resulting in the destruction of liver cells, causing inflammation, liver failure, and fibrosis. Pediatric (AIH) is an autoimmune inflammatory disease that usually requires immunosuppression for an extended period. Frequent relapses after treatment discontinuation demonstrate that current therapies do not control intrahepatic immune processes. This study describes targeted proteomic profiling data in patients with AIH and controls. A total of 92 inflammatory and 92 cardiometabolic plasma markers were assessed for (i) pediatric AIH versus controls, (ii) AIH type 1 versus type 2, (iii) AIH and AIH–autoimmune sclerosing cholangitis overlapping syndrome and (iv) correlations with circulating vitamin D levels in AIH. A total of 16 proteins showed a nominally significant differential abundance in pediatric patients with AIH compared to controls. No clustering of AIH subphenotypes based on all protein data was observed, and no significant correlation of vitamin D levels was observed for the identified proteins. The proteins that showed variable expression include CA1, CA3, GAS6, FCGR2A, 4E-BP1 and CCL19, which may serve as potential biomarkers for patients with AIH. CX3CL1, CXCL10, CCL23, CSF1 and CCL19 showed homology to one another and may be coexpressed in AIH. CXCL10 seems to be the central intermediary link for the listed proteins. These proteins were involved in relevant mechanistic pathways for liver diseases and immune processes in AIH pathogenesis. This is the first report on the proteomic profile of pediatric AIH. The identified markers could potentially lead to new diagnostic and therapeutic tools. Nevertheless, considering the complex pathogenesis of AIH, more extensive studies are warranted to replicate and validate the present study’s findings.

## 1. Introduction

Autoimmune hepatitis (AIH) is characterized by immune-mediated hepatocyte injury resulting in the destruction of liver cells, causing inflammation, liver failure, and fibrosis [1]. (AIH) is characterized by a female preponderance [1], circulating autoantibodies, hypergammaglobulinemia (IgG) and histological findings that describe a dense lymphoplasmocytic infiltrate of the portal tract suggesting interface hepatitis [2,3,4,5,6]. Interface hepatitis constitutes hepatocyte apoptosis [7,8,9] that can transform hepatocytes into myofibroblasts, progressing to liver fibrosis [10,11]. The inflammatory infiltrate can expand into the liver parenchyma and form portal bridges with lobular collapse [12,13]. The clinical severity, histological manifestations and disease outcome result from the outstanding cellular and molecular disease mechanisms [4]. Autoimmune sclerosing cholangitis (ASC) is a progressive liver disease characterized by intrahepatic and/or extrahepatic biliary tree inflammation with elusive etiology, resulting in bile duct injury and liver fibrosis. The overlapping syndrome between AIH and ASC has been more frequently described in children than adults since the increasing usage of noninvasive biliary imaging [14]. Autoantibodies are the serologic hallmark of AIH. The presence of antinuclear antibody (ANA) and/or antismooth muscle antibodies (SMA) indicates AIH type 1 (AIH-1), whereas antiliver kidney microsomal antibody type one (LKM-1) and/or antiliver cytosol type one antibody (LC-1) are attributed to AIH type 2 (AIH-2) [14]. Usually detected with routine immunofluorescence testing, autoantibodies are not specific for AIH and can also be detected due to molecular mimicry in a small percentage of patients with viral hepatitis [15], drug-induced liver injury or other autoimmune disorders [16]. Therefore, specific biomarkers are further needed for the diagnosis of AIH. In addition to the potential diagnostic application, novel AIH autoantigens could also contribute to a better understanding of the pathogenesis of the disease. Although various AIH target autoantigens have been discovered and described, little is known about their pathogenetic role. Pathogenetic autoantibodies must be expressed on target cells, either on the plasma membrane or secreted by the cells. They should be binding to the specific antigen to disturb a cellular function directly or indirectly [17]. These criteria are met by specific autoantibodies for cytochrome P450 2D6 (CYP2D6) or Asialoglycoprotein receptor 1 (AGPR-1) or sulfated glycosaminoglycan, being both present on the liver cell membrane [18].

Additionally, chemokines are small proteins present in the damaged tissue, which can influence the immune and inflammatory cells’ migration to the liver and contribute to hepatic fibrosis [19,20,21,22,23,24,25,26,27,28,29]. Chemokines induce chemotactic activity in injured hepatocytes, hepatic stellate cells, endothelial cells and dendritic cells [30]. The chemokine receptors are mostly found on the immune and inflammatory cells [30]. Effector cells of the same family can attract various ligands that express the same receptor, enabling an excessive immune response [22,30,31]. The chemokine ligands cited in immune-mediated liver diseases are CXC motif chemokine ligands (CXCL) 9–10, 12 or 26. Chemokine differences could correlate with inflammatory activity and disease severity and could be used as potential biomarkers or therapeutic targets [11].

A significant increase in the number of proteomic studies in recent years has led to essential data regarding the implication of proteins in autoimmune diseases, such as osteoarthritis, rheumatoid arthritis, psoriatic arthritis, ankylosing spondylitis, systemic lupus erythematosus, systemic sclerosis and Sjogren’s syndrome [32].

Several studies identified candidate biomarkers for AIH with proteomics tools. For example, patients with AIH-1 presented a heterogenous nuclear ribonucleoprotein A2/B1 (hnRNP-A2/B1) [33] and liver arginase, HSP60, HSP70, HSP90 and valosin-containing protein [34,35]. Fumarate hydratase and phosphoglycerate mutase isozyme B were described in Chinese patients with AIH [36,37], and other autoantigens were mentioned in various studies (IL4R, AL137145, LOC646100, C17orf99, METRNL, APCDD1L [17], lamin, histone, cyclin A and U1RNP-A [38]).

Vitamin D intervenes in the etiopathogenesis of AIH by inhibiting the activation of elevated levels of toll-like receptors (TLRs) -2, TLR-4 and TLR-9 [38]. Clinical studies often describe the association between low vitamin D levels and increased severity of interface hepatitis and liver fibrosis [39]. *VDR* and *CTLA-4* alleles are involved in the immune process of AIH [40]. The detection of low vitamin D levels in nonresponders to glucocorticoid therapy has led to correlations between AIH and vitamin D as a possible prognostic biomarker [39].

However, proteomic analysis has not previously been performed in pediatric patients with AIH. To address this important research gap, the main objective of our study was to identify whether patients with AIH differ in their inflammatory status by assessing their proteomic profile and levels of vitamin D. We assessed the serum markers using two 92-target panels (Inflammation and Cardiometabolic panels, Olink, Uppsala, Sweden) in patients with AIH and healthy control samples. Further, the aim of our study was to identify the differences between patients. In this regard, we compared generated protein profiles to obtain (i) a comparison between the types of AIH, (ii) a comparison between AIH and overlapping syndrome AIH–ASC and (iii) correlations between AIH and vitamin D as a possible prognostic biomarker. Our study is the first one to describe the serum proteome profiles of patients with AIH. Subject to replication and validation, the identified markers could potentially lead to new diagnostic and therapeutic tools.

## 2. Results

### 2.1. Clinical Cohort

The study groups included in the current study comprised 20 pediatric patients with AIH and 19 healthy controls. Patients with concomitant liver diseases, drug-induced liver injury, acute or chronic infections and those with de novo AIH who had undergone liver transplantation were excluded. Table 1 depicts the patients’ clinical and serological features. The controls were age- and sex-matched patients with no acute or chronic liver diseases and no acute or chronic infections with no systemic autoimmune diseases (Figure 1a,b). The median level of the total vitamin D in the patients with AIH was 15 ng/mL (Interquartile range IQR, 9–22 ng/mL), and it was significantly lower in the serum samples from the patients with AIH than from the age- and sex-matched healthy controls (15 ng/mL vs. 47 ng/mL, *p* < 0.0001) (Figure 1c). Alanine transaminase, alkaline phosphatase, gamma-glutamyl transpeptidase and total bilirubin levels were significantly negatively correlated with vitamin D levels, while no association was observed between serum vitamin D levels and aspartate transferase, albumin, γ-Globulin levels and platelet count (Table 2).

### 2.2. Targeted Proteome Profiling

Principal component analysis (PCA) of the remaining 18 healthy controls and 18 patients is shown in Figure 2 for the Cardiometabolic panel and Figure 3 for the Inflammation panel. No clear separation of patients and controls was observed when considering all proteins. However, two patients with AIH were significantly distinct from the rest of the group on PC2 and PC3 consistently between the two proteome panels. These samples correspond to two patients with AIH presenting with decompensated cirrhosis. The remaining patients were in remission, and the serum samples were collected during follow-up.

### 2.3. Comparison of Protein Abundance in Patients with AIH and Controls

In order to assess differentially expressed serum markers in the patients with AIH compared to the controls, protein abundance was compared between the healthy controls and patients with AIH using group *t*-test analysis. Nominally significant differences in protein abundance were identified, such as CA1, CA3, GAS6, FCGR2A, TIMD4 and EFEMP1, in the Cardiometabolic panel (Figure 4a) or 4E-BP1, CCL19, CSF-1, CX3CL1, CCL23, IL-18R1, IL-10RB, OPG, CXCL10 and CDCP1 in the Inflammation panel (Figure 4b). However, these differences did not reach statistical significance at a p-adjusted level. The two proteome panels’ top candidate proteins that were down-regulated or up-regulated are also individually depicted in Figure 5.

### 2.4. Comparison of Proteome Profiles between AIH Subphenotypes

Next, we assessed whether proteome profiles could discriminate between subphenotypes of pediatric AIH. We analyzed all protein data within the group to assess clustering based on AIH type 1 (AIH-1) (twelve patients) or type 2 (AIH-2) (six patients). The results suggest that the samples are rather heterogeneous and do not provide a clear clustering by type of AIH (Figure 6). We also compared the proteome profiles in patients with AIH (ten patients) and with overlap ASC (eight patients), and the results did not show the clustering of samples based on these conditions (Figure 6).

### 2.5. Correlations between AIH and Vitamin D

Since vitamin D deficiency is a well-known feature in AIH [39], and vitamin D is also known to exert potent immunomodulatory roles [40,41,42], we also assessed the association of serum vitamin D levels with the differentially expressed proteins in the AIH group identified in the Cardiometabolic panel (CA1, CA3, GAS6, FCGR2A, TIMD4, EFEMP1) and Inflammation panel (4E-BP1, CCL19, CSF-1, CX3CL1, CCL23, IL-18R1, IL-10RB, OPG, CXCL10). No clear correlations were observed between these proteins and circulating vitamin D (Figure 7a,b).

## 3. Discussion

An important step towards improving the management of AIH is to contribute to a better understanding of disease etiopathogenesis. Without early and adequate treatment, the chronic process of AIH can advance to cirrhosis and liver failure, significantly impairing the quality of life. Approximately one-third of adult and one-half of pediatric patients present cirrhosis at diagnosis [43]. To avoid the final stage of disease complications, early diagnosis is necessary to properly assess the risk factors of illness progression. Lack of the means for early diagnosis is the present problem in preventing disease complications.

We applied proteome analysis to blood specimens of pediatric patients with AIH and healthy controls. PCA on proteome data did not reveal sample segregation based on case or control status. Nevertheless, we found 16 proteins that showed nominally significant differential expression in the circulation of patients with AIH, namely CA1, CA3, GAS6, FCGR2A, TIMD4, EFEMP1, CSF-1, CX3CL1, CCL23, IL-18R1, IL-10RB, OPG, 4E-BP1, CXCL10 and CCL19. These proteins were differently expressed among patients with AIH and controls with up-regulated proteins (GAS 6, FCGR2A, CXCL10 and CCL19) and down-regulated proteins (CA1, CA3 and 4E-BP1). Based on the observed patterns, these identified proteins could be potential new biomarkers for AIH.

However, these proteomic profiles did not differentiate patients based on type 1 or type 2, AIH, or based on AIH vs. AIH–ASC. This relatively small number of differentially expressed proteins and the slight difference between disease subphenotypes could be due to several limitations: a relatively small sample size, sample heterogeneity and the fact that most serum samples were collected from patients with AIH after the initiation of therapy at follow-up after years of the controlled disease.

The advent of affinity-based proteomic technologies highlights the importance of biomarker discovery [44]. The PEA developed with Olink Proteomics has received increasing attention and surpassed the studies presenting MS-based approaches. The magnitude of highly specific antibodies and primers places substantial sensitivity and specificity for assays in biological samples [45]. The essence of the precision of PEA has been used in the pathogenesis of liver diseases. This proteomic analysis has not been performed before in pediatric patients with AIH.

There is limited information regarding PEA in AIH in other studies, most of them experimental studies. One study tested if complexed IL-2/anti-IL-2 in mice could increase the selectivity for intrahepatic regulatory T cells (Tregs) in the late course of AIH with the Olink mouse exploratory panel. Complexed IL-2/anti-IL-2 managed to stabilize the numbers of Tregs and intrahepatic effector T cells (Teffs) within the liver, resulting in AIH improvement [46]. The same Olink panel was used in splenectomized mice followed by induction of AIH. Splenectomized mice presented more severe portal inflammation. The results indicated that the spleen does not contribute to AIH induction, and splenectomy interrupts the immune regulation by increasing IL-17, IL-23 receptors and caspase 3 that generate liver inflammation and apoptosis [47]. Furthermore, the same authors treated the experimental murine with anti-CD20 during the late stage of AIH. The results suggested that anti-CD20 therapy solely is ineffective in AIH [48].

Moreover, Olink proteome profiling was used in several liver diseases. One study identified the inflammatory modulator STAT3 and the E2 component of the mitochondrial pyruvate dehydrogenase complex (PDC-E2) in cholangiocytes and hepatocytes. Both proteins presented higher expression in cirrhotic primary biliary cholangitis (PBC) livers [49]. One study that included subjects with PBC from the UK Biobank discovered nineteen proteins with significant expression even in patients treated with ursodeoxycholic acid. Six proteins were tightly linked to chemokines, including C-C motif chemokine 20 (CCL20) [50]. The PEA technology was also used in multiple studies to describe the proteomic profile in viral hepatitis. Various inflammatory protein biomarkers were described in patients with liver transplants and active HCV infection, such as C-X-C motif chemokine 10 (CXCL10), CXCL11, C-C motif chemokine 19 (CCL19), CCL20, interferon γ, interleukin (IL) -18R1 and tumor necrosis factor-β [51]. The same chemokines were mentioned in patients with hepatitis C before and during treatment with glecaprevir/pibrentasvir (GLE/PIB) +/− ribavirin [52] and treatment with ledipasvir/sofosbuvir [53]. One large study analyzed the levels of 4907 plasma proteins in 35,559 Icelanders and 1459 proteins in 47,151 UK Biobank participants to identify proteins involved in nonalcoholic fatty liver disease (NAFLD) with Olink proteome profiling. The results led to eighteen sequence variants associated with NAFL and four with cirrhosis, meaning that proteomics can differentiate between NAFL and cirrhosis [54].

In our study, the differently expressed proteins among the patients with AIH and controls (GAS 6, FCGR2A, CXCL10, CCL19, CA1, CA3 and 4E-BP1) are described in multiple liver diseases with proteomic analysis that uses different technologies, such as two-dimensional gel electrophoresis (2-DE), image analysis and mass spectrometry (MS). GAS6 protein was up-regulated in our patients with AIH. The TAM (Tyro3, AXL, Mer) receptor ligand GAS6 is primarily expressed by Kupffer cells and is a vitamin K-dependent protein with a high affinity for the AXL receptor [55,56,57]. The GAS6/AXL pathway promotes HSC activation and could attenuate hepatic fibrosis [55]. Clinical trials have demonstrated elevated GAS6 and AXL levels in patients with chronic hepatitis C (HCV) and acute liver disease (ALD) [55]. One recent prospective study including 154 patients undergoing liver resection evaluated soluble AXL (sAXL) and GAS6 in the immediate perioperative and postoperative periods. The GAS6/AXL pathway was expressed in the case of underlying liver disease, and its inhibition appeared after the induction of liver regeneration, resulting in immune activation [58]. Their implication in autoimmune phenotypes is underlined by an experimental study on mice knockout for all three TAM receptors (TAM triple knockout; TAM TKO) that developed a spontaneous liver disease that resembles AIH [59]. The association between GAS6 and liver fibrosis and autoimmune phenotypes could explain the difference between the AIH cohort and controls regarding GAS6 expression.

Another up-regulated protein in our patients with AIH is FCGR2A, which presents low-affinity Fcγ receptors (FCGRs) that intervene in immunoglobulin G (IgG) antibody effects on leukocytes. FCGRs are involved in phagocytosis, recruitment to inflammatory lesions, antibody-dependent cellular cytotoxicity, regulation of B-cell activation and release of inflammatory mediators [60]. When inappropriately activated, the same mechanism also results in the development of autoimmune diseases [61]. The Fc receptor locus presents marked genetic variability, leading to an increased risk of developing autoimmune diseases and impacting the defense against infection [62,63,64,65,66,67]. Only one clinical study has genotyped FcγRIIA, FcγRIIB and four Fc receptor-like gene 3 (FCRL3) polymorphisms in 87 Japanese patients with type 1 AIH and matched controls. Still, they observed no difference in the distribution of the genotypes between patients and controls, suggesting that type 1 AIH was not influenced by FcγRIIA, FcγRIIB or FCRL3 polymorphisms [68]. There is scarce information regarding the role of FCGR2A in AIH, and further research is needed for this association. In light of those mentioned above, the overexpression of FCGR2A in our patients with AIH renders it an interesting target for future studies.

CCL19 is a proinflammatory human chemokine protein of the intercrine beta family [Cystein–Cysteine Chemokine] [69,70,71], and it is up-regulated in patients with AIH versus controls. This chemokine plays an important role in autoimmune diseases and chronic inflammatory disorders [72,73]. The CCR7/CCL19/CCL21 axis establishes interactions between antigen-presenting cells (APCs) and antigen-specific lymphocytes, representing a key process in adaptive immune system function [74]. Neo-lymphoid follicles expressing CCL19 and CCL21 are encountered in chronic inflammatory liver diseases, such as PBC, PSC and chronic hepatitis C [75]. They are involved in liver lymphocyte recruitment and maintaining the chronic inflammatory infiltrate [76]. Some studies link CCL19 with inflammation in autoimmune diseases, but there is limited information regarding its role in AIH. Targeting the CCL19/CCR7 complex, necessary for lymphocyte trafficking and also overexpressed in our AIH cohort, could represent a novel anti-inflammatory drug discovery approach [69].

CXCL10 was up-regulated in our patients with AIH versus the controls. Elevated serum levels of CXCL10 are described in patients with AIH in correspondence with liver inflammation, primary biliary cirrhosis (PBC), chronic viral hepatitis [77], mixed cryoglobulinemia and autoimmune thyroiditis [78]. CXCL10 was proposed as a biomarker in progressive fibrosis in African-American patients with chronic hepatitis C [79], and its levels were inversely correlated with the prognosis of interferon therapy [80]. Being present in various inflammatory liver diseases, CXCL10 could be an important therapeutic agent [27,30].

Bile duct epithelium contains an important amount of carbonic anhydrase. Antibodies to this enzyme are disease-specific markers of injury to the biliary epithelium, often encountered in autoimmune disorders [81]. Patients with ASC are different from patients with PBC by presenting more elevated serum levels of aspartate transaminase (AST) and lower serum levels of immunoglobulin M [82], and they are also characterized by the presence of antibodies to carbonic anhydrase in serum more often than patients with PBC or AIH [81,83]. CA1 and CA3 proteins were found to be down-regulated in our patients with AIH, and CA3 was described in various systemic autoimmune diseases [84]. Even though carbonic anhydrase is abundant in bile duct epithelium, the role of CA1 and CA3 in AIH is not known and has not been described in detail.

There is scarce information regarding the role of 4E-BP1 in AIH, and in our study, the levels of 4E-BP1 were lower in the patients with AIH compared to the healthy controls. In experimental studies, 4E-BP protein synthesis is promoted by the mechanistic target of rapamycin complex 1 (mTORC1), which is involved in regulating cell growth and metabolism [85]. The mechanistic target of rapamycin (mTOR) participates in autophagy regulation that can be involved in liver injury [86]. mTOR plays a key role in innate and adaptive immune responses [87]. Phosphatidylinositol 3 Kinase PI3K/AKT/mTOR-mediated autophagy was demonstrated to play a role in reversing liver fibrosis [86].

The prevalence of AIH has been rising in the past two decades, especially in young patients under 20 years old and in patients with ages between 50 and 69 years [88]. AIH has typical serological and genetic characteristics regarding each age group [89]. It would be of great interest to compare the proteomic analysis among pediatric, adult and elderly patients with AIH. There are several studies that enrolled adult patients with AIH and have identified candidate biomarkers for AIH with proteomics tools. One study identified heterogenous nuclear ribonucleoprotein A2/B1 (hnRNP-A2/B1) [33], liver arginase, HSP60, HSP70, HSP90 and valosin-containing protein [34,35] in adult patients with AIH-1. Other autoantigens are mentioned in various studies, which include adult patients with AIH, such as fumarate hydratase and phosphoglycerate mutase isozyme B [36,37], and IL4R, AL137145, LOC646100, C17orf99, METRNL, APCDD1L [17], lamin, histone, cyclin A and U1RNP-A [38]. Based on the proteins that showed nominally significant differential expression in the circulation of our pediatric patients with AIH, some were different from those cited in studies that included adult AIH, and some were described in other chronic liver diseases as mentioned above. Further research could provide more insight regarding the differences between age groups.

In the present study, the patients with AIH presented 25(OH)D deficiency compared to the controls. Analyzing the expressed proteins and the serum level of vitamin D, we found no significant correlations in both Olink panels. The small number of pediatric patients with AIH could be the cause of these results.

Vitamin D deficiency is frequently encountered in AIH [39,40,90,91]. In AIH, an increased number of monocytes express a high level of regulated intracellular toll-like receptors (TLRs). Vitamin D inhibits TLR-2, TLR-4 and TLR-9 activation [39,92]. The non-genomic role of vitamin D in AIH comprises the up-regulation of the phosphatase 1 mitogen-activated protein kinase (MAPK) signaling pathways, controlling cytokine production. 25(OH)D inhibits Gamma delta (γδ) T cells, a small subset of T cells described in proinflammatory reactions [92,93]. 25(OH)D intervenes against oxidative injuries resulting in nitrite production, decreases lipid peroxidation and stimulates the hepatic antioxidant system [92,93]. Clinical trials often describe the association between decreased levels of 25(OH)D and increased severity of interface hepatitis and important liver fibrosis [94,95]. Some *VDR* and *CTLA4* variants are involved in developing the immune process as part of AIH pathogenesis. The genotype variants are connected to fatty acid synthase (FAS) promoter variants or other TNF superfamily proinflammatory cytokines. This mechanism presents great potential in discovering disease-specific liver fibrosis [96,97]. Decreased circulating levels of 25(OH)D are reported in 81% of Turkish patients with AIH compared to healthy individuals. These patients are also more often nonresponders to glucocorticoid therapy than patients with AIH without 25(OH)D deficiency [94]. In this regard, 25(OH)D deficiency has been proposed as a prognostic biomarker in AIH [94,95].

In summary, in this pilot study, we describe for the first time the proteomic profile of pediatric AIH compared to controls. We show differences in circulating levels of proteins that, based on existing literature, are relevant to AIH pathogenesis. Future studies in larger cohorts and newly diagnosed patients are warranted to validate the findings revealed by the current study and assess the performance of these biomarkers in the diagnosis of AIH. Despite the current treatments (prednisone and azathioprine) and the immunosuppressive response of these drugs, little is mentioned about the metabolic reactions that occur in vivo, which intervene with the autoimmune response and inflammation. Further research into this interaction could deliver a new perspective on the pathogenesis of autoimmune liver diseases and novel therapeutic agents. Our data further show the serological heterogeneity in pediatric AIH and propose a variety of mechanisms underlying AIH. Identifying new specific autoantigens in AIH may influence characterizations of the autoimmune responses and explorations of their pathogenic role. Further investigations of autoantibodies in pediatric AIH may decide their value for diagnosis and gain insight into the pathogenesis in pediatric AIH.

## 4. Materials and Methods

### 4.1. Patients

We enrolled 39 participants, pediatric AIH (*n* = 20) and controls (*n* = 19). Pediatric patients fulfilled the diagnostic simplified criteria for AIH defined by the International Autoimmune Hepatitis Group [98]. The patients included in our study were enrolled between February 2021 and March 2022. The simplified AIH scoring system includes the presence of autoantibodies, immunoglobulin G, histology and exclusion of viral hepatitis [99]. AIH-1 was defined by the presence of ASMA and/or ANA and AIH-2 by the presence of LKM-1 or LC-1 antibodies. Patients with de novo AIH occurring after liver transplant (LT) and other concomitant liver diseases (viral, metabolic, Wilson’s disease, toxic causes) were excluded. Based on the simplified AIH criteria [98], 7 patients had a score of ≥7 (definite AIH), and 13 patients had a score of ≥6 (probable AIH). Acute liver failure was defined as an INR (international normalized ratio) >2 and encephalopathy within 8 weeks of diagnosis. The overlap syndrome of AIH and sclerosing cholangitis (AIH–SC) was defined by typical cholangio-magnetic resonance imaging (cholangio-RM) features. Cirrhotic AIH was defined based on laboratory analysis and liver stiffness on transient elastography (FibroScan, Echosens, Paris, France), equivalent to stage F4 METAVIR. Remission was defined as the normalization of aminotransferase and IgG levels. All patients were started on steroids with the addition of a second agent depending on the response to steroids. Azathioprine metabolite monitoring was not universally performed. Children were categorized as AIH-1 versus AIH-2, acute liver failure versus cirrhotic AIH and overlap sclerosing cholangitis versus nonoverlap. Controls were obtained from age- and sex-matched patients with no acute or chronic liver diseases or systemic autoimmune diseases. We provided ranges for the median values and standard deviations for the mean values. The Institutional Review Board approved the study protocol, and all participants provided informed voluntary consent.

### 4.2. Sample Processing and Protein Detection

Serum samples were collected from the 39 participants and stored at −80 °C until shipment. The serum samples were shipped for protein profiling to *Radboud University Medical Center*, Nijmegen, The Netherlands, using cold chains. A total of 184 proteins were measured in the serum using the Olink Cardiometabolic and Inflammation panels (92 proteins each) (Olink, Uppsala, Sweden). Proteins were quantified with the proximity extension assay (PEA). Data were reported in Normalized Protein eXpression values (NPX), which are calculated on a log_2_ scale [100]. One sample attributed to one patient in the Inflammation panel was flagged as “Warning” by Olink and was further excluded from the analysis. The analysis within the Inflammation panel was performed on 19 patients with AIH and 19 controls. Regarding the Cardiometabolic panel, two samples, one patient and one control, fell outside +/−3 standard deviations from the mean IQR and +/−3 standard deviations from the mean sample median with the olink_qc_plot function in the Olink^®^ Analyze library and were considered outliers and, therefore, removed from further analysis. Subsequent analysis within the Cardiometabolic panel was carried out in the remaining 19 patients and 18 control samples. Several proteins in both panels measured at or below the lower limit of detection (LLOD) in more than 50% of the samples were filtered to be removed from further analyses. For the remainder of the proteins, levels measured at or below LLOD were used as such. Seventy-eight proteins from the Cardiometabolic panel and 63 from the Inflammation panel further proceeded into unsupervised learning and differential expression analyses. After sample QC, two patient samples and one control were excluded from further analysis. Data were intensity normalized (v.2) before analysis.

### 4.3. Statistical Analysis

To observe patterns in our data set, we used principal component analysis (PCA). Heatmaps were also used to visualize clusters of samples or features. The significance of the difference between healthy controls and patients with AIH across the entire set of serum proteomics readouts was assessed using the olink_ttest function in the Olink^®^ Analyze library in R, which performs a Welch 2-sample *t*-test at a confidence level of 0.95 for every protein using the function t.test from the R library stats and corrects for multiple testing using the Benjamini–Hochberg method (“FDR”) using the function p.adjust from the R library stats. A group *t*-test was performed on AIH–Control. Box plots were used to visualize the distribution and difference between healthy controls and patients with AIH for a given targeted protein. Associations between serum protein levels measured by Olink and vitamin D for the AIH-Control differentially expressed proteins were evaluated with Pearson correlation. Unsupervised learning and differential expression analyses were performed in R using Olink^®^ Analyze. R version: R 4.2.1 with RStudio (R Core Team (2023). R: A language and environment for statistical computing. R Foundation for Statistical Computing, Vienna, Austria) as an IDE for R. Base packages: Olink^®^ Analyze, xlsx, stats, dplyr, ggplot2, cowplot. Running under Windows 10 x64, 22H2 (build 19045).

## Figures and Tables

**Figure 1 ijms-24-07479-f001:**
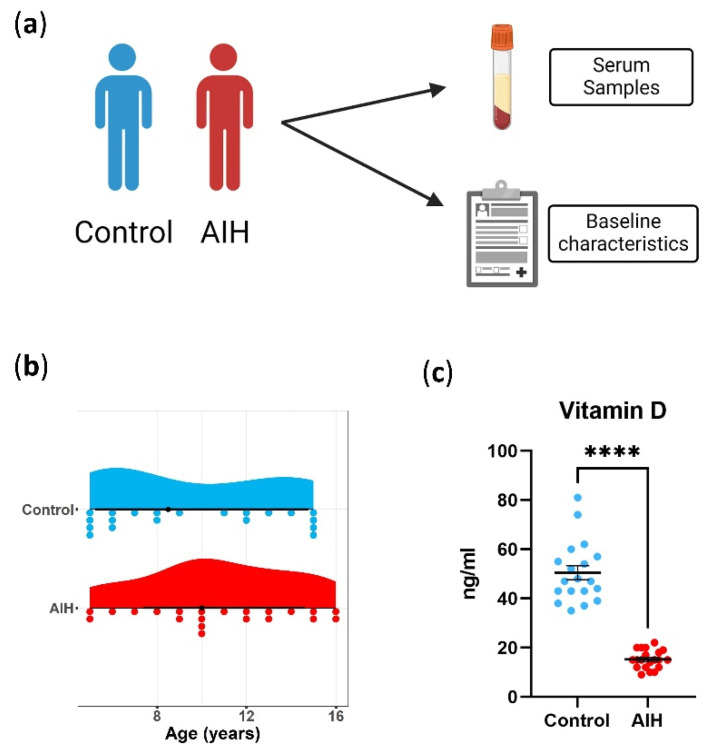
Baseline characteristics of patients. (**a**) Schematic representation for data collection (healthy controls in blue and patients with AIH in red). (**b**) Age in all patients included in the study. (**c**) Vitamin D in serum. Each dot represents measurements from one patient; unpaired *t*-test with data presented as the mean with SEM, *p* < 0.0001 (****).

**Figure 2 ijms-24-07479-f002:**
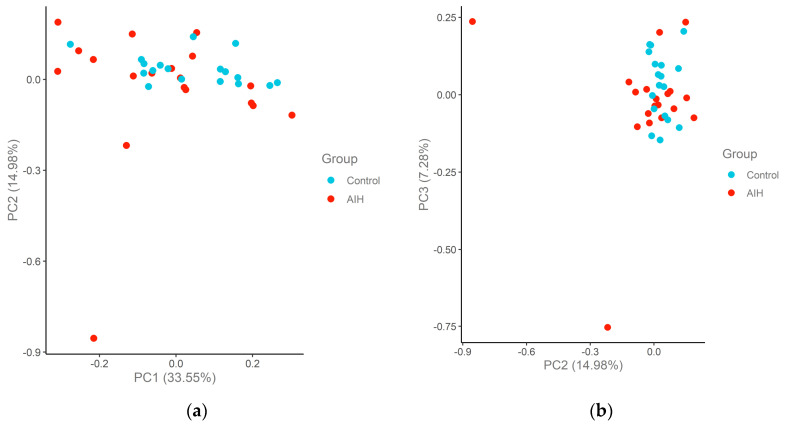
Principal component analysis (PCA) plots of the AIH and control cohort samples in the Cardiometabolic panel. The samples are colored according to the group. The percentage of variation that accounted for each principal component is shown in brackets along the x and y axis label. (**a**) PC1–PC2; (**b**) PC2–PC3.

**Figure 3 ijms-24-07479-f003:**
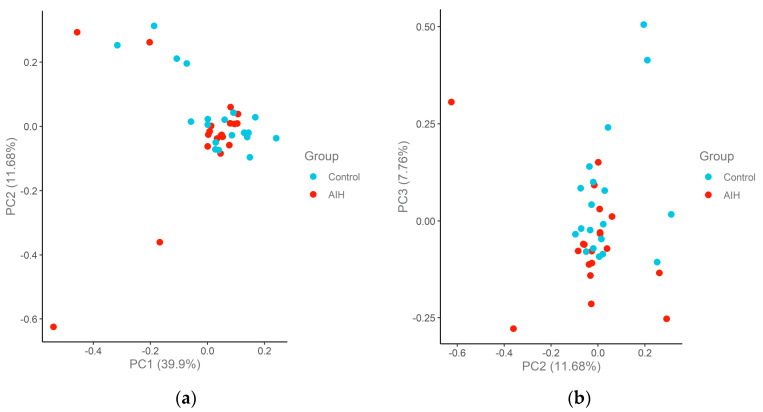
Principal component analysis (PCA) plots of the AIH and control cohort samples in the Inflammation panel. The samples are colored according to the group. The percentage of variation that accounted for each principal component is shown in brackets along the x and y axis label. (**a**) PC1–PC2; (**b**) PC2–PC3.

**Figure 4 ijms-24-07479-f004:**
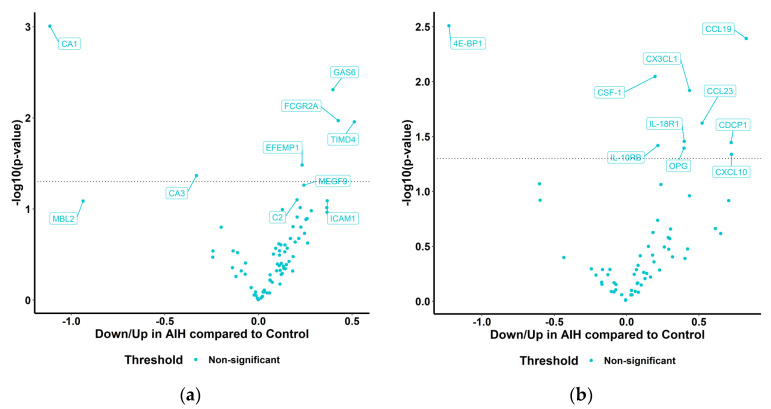
Volcano plots representing differentially abundant proteins (DAP) in AIH compared to controls in the Cardiometabolic and Inflammation panels. Welch 2—sample *t*-test or paired *t*-test at a confidence level of 0.95 for every protein; *p* value shown as logarithm in base 10 on the y-axis. (**a**) Cardiometabolic and (**b**) Inflammation panels.

**Figure 5 ijms-24-07479-f005:**
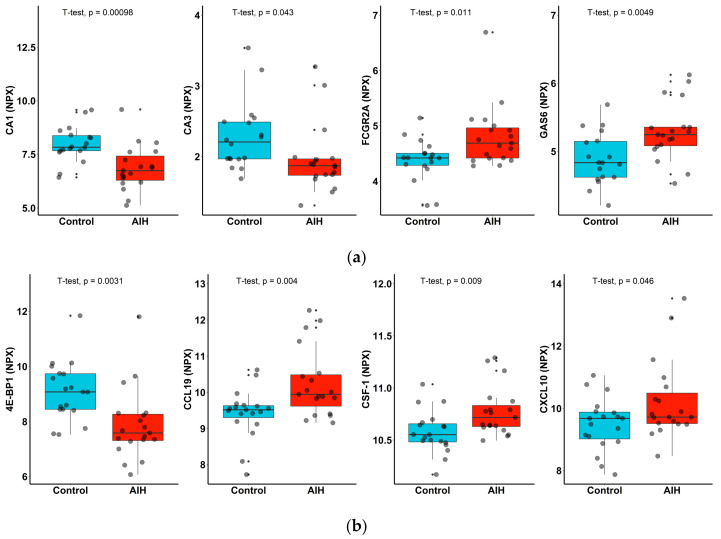
Levels of serum proinflammatory markers in patients with AIH (red) and controls (blue). Boxplots of up-regulated (GAS 6, FCGR2A, CXCL10 and CCL19) and down-regulated proteins (CA1, CA3 and 4E-BP1) in the (**a**) Cardiometabolic and (**b**) Inflammation panels. Un-paired *t*-test with corresponding *p* values are shown on plots for each protein.

**Figure 6 ijms-24-07479-f006:**
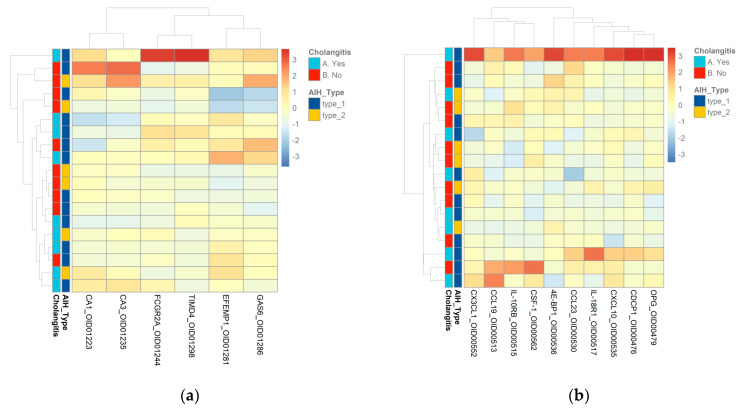
(**a**) Heatmap of AIH type 1 and AIH type 2 samples and AIH–ASC samples in the Cardiometabolic panel. (**b**) Heatmap of AIH type 1 and AIH type 2 samples and AIH–ASC samples in the Inflammation panel. Columns represent individual proteins, while rows represent the samples. Row and column dendrograms show the distance/similarity between the variables. The relative value for each protein is shown by the color intensity of the Z score with red indicating up-regulated and blue indicating down-regulated proteins.

**Figure 7 ijms-24-07479-f007:**
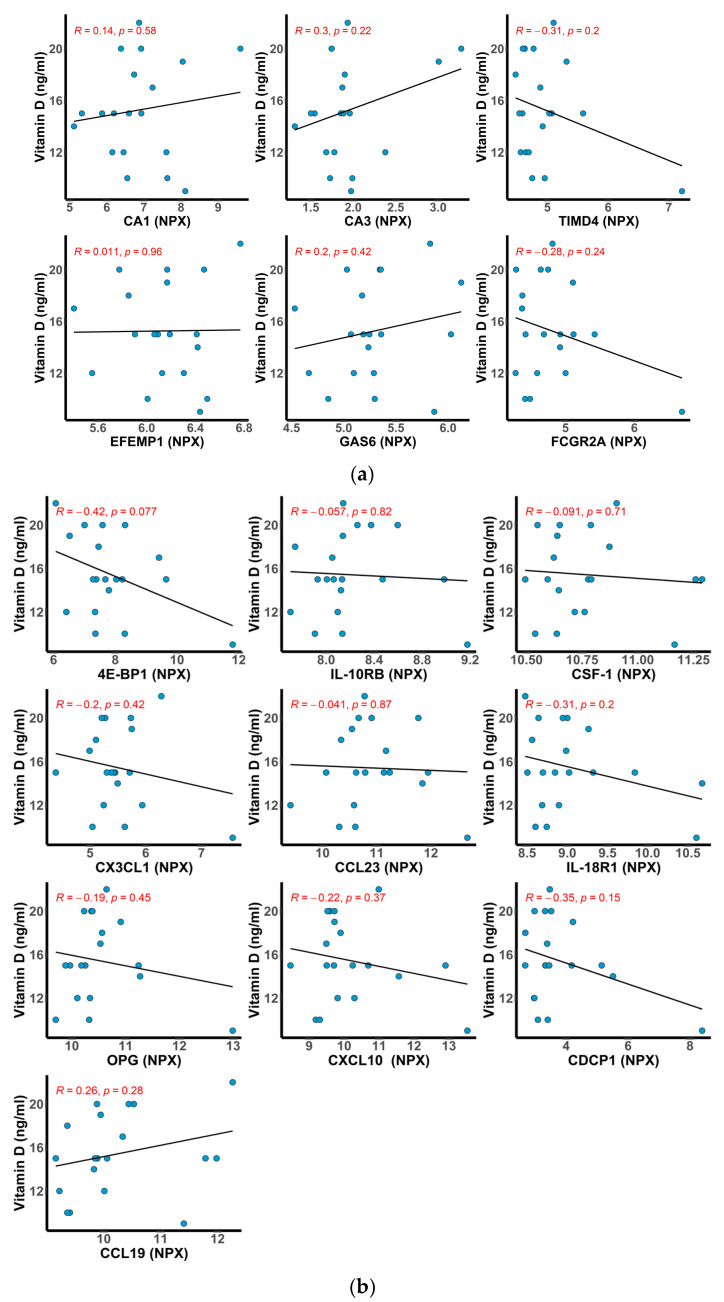
Scatter plots showing the correlation (Pearson) between the top differentially abundant proteins in AIH and the levels of serum vitamin D. The fitted linear regression line of each scatter plot is depicted along with the *R* and *p* values. (**a**) Cardiometabolic and (**b**) Inflammation.

**Table 1 ijms-24-07479-t001:** Clinical and serological features of patients with AIH tested for autoantibodies.

Characteristics	AIH (*n* = 20)
Age (median; range) (yr)	9.95; 2.5–15.8
Sex (male/female)	5/15
ALT (U/L; median; range)	251.5; 104–2988
AST (U/L; median; range)	184.5; 110–2441
ALP (U/L; median; range)	291.5; 86–1620
GGT (U/L; median; range)	74.5; 6–574
TB (mg/dL; median; range)	1.28; 0.3–6.5
γ-Globulin (g/L; median; range)	1902; 467–2553
ALB (g/dL; median; range)	4.3; 3–4.9
PLT (×10⁴/µL; median; range)	27; 12–57
ANA Positive (%)	85%
SMA Positive (%)	25%
LKM-1 Positive (%)	10%
Lc-1 Positive (%)	10%
Anti-SLA positive (%)	5%

Abbreviations: AIH, autoimmune hepatitis; ALT, alanine transaminase; AST, aspartate transferase; ALP, alkaline phosphatase; GGT, gamma-glutamyl transpeptidase; TB, total bilirubin; ALB, albumin; PLT, platelet count; ANA, antinuclear antibody; LKM-1, liver kidney microsomal type 1 antibody; Lc-1, antibodies to liver cytosol antigen type 1; SMA, antismooth muscle antibody with anti-actin cable specificity; SLA, soluble liver antigen.

**Table 2 ijms-24-07479-t002:** Relationship between vitamin D and laboratory parameters in patients with AIH.

Variable	Vitamin D	
	*r*	*p*
ALT (U/L)	−0.5117	0.0211 *
AST (U/L)	−0.4331	0.0564
ALP (U/L)	−0.6470	0.0020 *
GGT (U/L)	−0.5291	0.0164 *
TB (mg/dL)	−0.4855	0.0300 *
γ-Globulin (g/L)	0.1595	0.5018
ALB (g/dL)	0.4144	0.0693
PLT (×10⁴/µL)	0.2796	0.2325

Abbreviations: ALT, alanine transaminase; AST, aspartate transferase; ALP, alkaline phosphatase; GGT, gamma-glutamyl transpeptidase; TB, total bilirubin; ALB, albumin; PLT, platelet count; *r*, Spearman correlation coefficient; * *p* value of <0.05 was considered significant.

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
