# Peer review of "Detection of Novel Biomarkers in Pediatric Autoimmune Hepatitis by Proteomic Profiling"

_ijms, 2023, doi:10.3390/ijms24087479_

Round 1
Reviewer 1 Report
In this study, the authors describe targeted proteomic profiling data in patients with autoimmune hepatitis (AIH) by applying proteome analysis to blood specimens of 20 AIH pediatric patients and 19 healthy controls. They found 16 proteins, namely CA1, CA3, 213 GAS6, FCGR2A, TIMD4, EFEMP1, CSF-1, CX3CL1, CCL23, IL-18R1, IL-10RB, OPG, 4E- 214 BP1, CXCL10 and CCL19, that showed nominally significant differential abundance in pediatric AIH patients compared to controls. No clustering of AIH subphenotypes based on all protein data was observed, and no significant correlation of vitamin D levels was seen for the identified proteins. Proteins that showed variable expression were CA1, CA3, GAS6, FCGR2A, 4E-BP1, and CCL19 which might serve as potential biomarkers for patients with AIH. CX3CL1, CXCL10, CCL23, CSF1, and CCL19 showed homology to one another and may be co-expressed in AIH. They concluded that their study Identified markers that could potentially lead to new diagnostic and therapeutic tools.
The study is of interest with a potentially novel approach to better characterized AIH patients. However, some relevant points deserve further details and should be addressed.
-Patients: please describe in detail how AIH diagnosis was performed as the authors stated "Pediatric patients fulfilled the diagnostic criteria for AIH defined by the International Autoimmune Hepatitis Group [98]". However, reference 98 refers to the Simplified AIH scoring system and this study focuses on pediatric patients. It would be of clinical relevance to stress that simplified AIH diagnostic criteria were also externally validated, as previously reported (Validation of simplified diagnostic criteria for autoimmune hepatitis in Italian patients. Hepatology. 2009 May;49(5):1782-3; ).
-Discussion: since AIH onset may be different and it is well known that it can onset in all decades, I would suggest further discussing how this proteomic approach could be also used to compare pediatric vs adult and elderly as AIH not rarely onset in elderly patients as previously reported (Clinical features of type 1 autoimmune hepatitis in elderly Italian patients. Aliment Pharmacol Ther. 2005 May 15;21(10):1273-7. doi: 10.1111/j.1365-2036.2005.02488.x).
-The last point worth mentioning is the importance of a new diagnostic approach as classical autoantibodies may have suboptimal specificity due to their potential presence in other settings such as HCV-related liver disease that is a well-known infection characterized by the occurrence of autoimmune phenomena such as autoantibodies (ANA and SMA) as well-described (HCV and autoimmunity. Curr Pharm Des. 2008;14(17):1678-85. doi: 10.2174/138161208784746824).
Reviewer 2 Report
My specific comments are mentioned below.
The introduction and discussion are poorly referenced with literature that does not support the statements made in the text.
The objectives of this study are absent. After the introduction of the associated research, one paragraph for the objectives statement is necessary.
The quality of futures is too low, please increase the quality and also increase the test size.
